# Autoencoder and Classifier based Joint-Guided Completion for Partial Multi-Modal Hashing

## Abstract

The Multi-Modal Hashing (MMH) method based on complete modalities cannot effectively handle incomplete multi-modal samples, thus requiring the completion of missing modalities. Existing completion methods typically use complete modality samples with the same label to generate completion information. On one hand, they cannot fully utilize the different information between samples with different labels; on the other hand, they cannot effectively extract the global structural information of multi-modal samples. Therefore, we propose the autoencoder and classifier based joint-guided completion for partial multi-modal hashing (JCPMH) method that integrates autoencoders and classifiers. First, to fully utilize the different information between samples with different labels, we design a multi-modal classification module composed of multiple classifiers to learn different information. Second, we concatenate the multi-modal data into a whole and extract cross-modal global structural information through an autoencoder. Finally, based on the hashing module, multi-modal classification module and autoencoder module, we design a loss function to guide the generator to generate more accurate completion information for learning hash codes. JCPMH can utilize partial multi-modal samples for offline training and handle incomplete multi-modal samples during online retrieval. Additionally, we conducted extensive experiments to demonstrate the effectiveness of this model.

## 1 Introduction

With the rapid growth of internet data, multi-modal hashing (MMH)Shen et al. (2015b); Lu et al. (2019; 2020); Zhu et al. (2020) has been widely applied to multi-modal retrieval tasks due to its low storage overhead and fast retrieval capabilities. However, most MMH works assume that the multi-modal samples being processed is complete, meaning that data from all corresponding modalities are present. This assumption limits the application of these methods. To address the issue of missing modalities, there are two approaches: one approach is to ignore the completion process for missing modalities and directly use partial modality dataZheng et al. (2021). However, this often leads to significant loss of multi-modal semantic information, especially when the partial data ratio (PDR)Zheng et al. (2021) is high. Therefore, we need to adopt the second approach, which involves completing the data before using the partial multi-modal samplesa, and then using the completed samples to learn hash codes.

Among the completion methods, NCHTan et al. (2023) uses a deep learning strategy, introducing neighbor information to dynamically generate completion information from complete multi-modal samples with the same label. On one hand, it cannot utilize information from partial modality data, resulting in low data utilization, and on the other hand, it ignores the discriminative information between samples of different categories. Additionally, existing methods often process data from multiple modalities separately to extract multi-modal semantic information, leading to the loss of global structural information of multi-modal samples. To solve these problems and effectively complete missing modalities, we propose a autoencoder and classifier based joint-guided completion for partial multi-modal hashing (JCPMH) method that integrates autoencoderHinton & Zemel (1993) and classifiers. Specifically, we concatenate data from multiple corresponding modalities into a whole and use an autoencoder to fully exploit the global structural information of multi-modal sam-

ples. Additionally, we design a multi-modal classification module composed of multiple classifiers to extract category-level discriminative information using label supervision. Unlike NCH, this classification module can extract information from partial modality data, improving data utilization and extracting richer information. We then use these two types of information to jointly guide the generation of completion information, feeding the completed and complete multi-modal samples into the subsequent hashing network to learn more accurate and discriminative hash codes. Both NCH and our JCPMH can handle incomplete multi-modal samples during offline training and online query stages. Our main contributions are as follows:

- We propose a partial multi-modal hashing method named JCPMH that effectively completes partial multi-modal samples to handle missing modalities during offline training and online query stages in multi-modal retrieval.

- To effectively complete missing modalities, on one hand, we extract global structural information from fully-paired samples. On the other hand, we learn discriminative information from all available data, including partial multi-modal samples. We then use both types of information to jointly guide the completion of missing modalities.

- We conducted extensive experiments to evaluate the model. Our method outperformed other existing models on public datasets. Additionally, we visualized the results of our model in completing missing data, demonstrating the effectiveness of our approach.

## 2 RELATED WORK

### 2.1 HASHING-BASED MULTI-MODAL RETRIEVAL

Different from single-modal hashingGionis et al. (1999); Gong & Lazebnik (2011); Shen et al. (2015a); Wang et al. (2018); Chen & Lu (2020); Zheng et al. (2020); Yu et al. (2022) and cross-modal hashingXu et al. (2017); Jiang & Li (2017), multi-modal hashing requires the fusion of multi-modal samples to learn compact binary codes. Considering the cost of manual annotation and the generality of methods, some unsupervised works Song et al. (2013); Liu et al. (2015); Shen et al. (2022; 2018) have been proposed. They do not require label information. Most of these unsupervised methods are based on shallow learning methods. They typically use graphs or matrices to construct relationships within or between modalities. Considering that the learned hash codes are discrete binary codes, they also introduce some discrete optimization strategies. Multiple Feature Hashing (MFH)Song et al. (2013) constructs affinity matrices for each modality internally and considers all these local structures in the subsequent optimization process to learn fused multi-modal representations. Multi-view Alignment Hashing (MAH)Liu et al. (2015) formulates regularized kernel non-negative matrix factorization to learn hash codes. It explores the semantic information and joint probability distribution of multi-modal data. Multi-view Discrete Hashing (MvDH)Shen et al. (2018) learns labels through spectral clustering and uses this supervised information to enhance the discrimination of the learned hash codes. Unsupervised multi-view distributed hashing (UMvDisH)Shen et al. (2022) proposes an unsupervised multi-modal distributed learning method that directly learns hash codes on multi-modal distributed data. It also introduces the alternating direction method of multipliers (ADMM)Boyd et al. (2011) to solve the decentralized sub-optimization problem.

Due to the lack of guidance from manually annotated labels, these unsupervised methods have limited ability to explore semantic information between samples. Supervised methodsLu et al. (2019; 2020); Zhu et al. (2020); Yang et al. (2017); Liu et al. (2020) focus on exploring semantic information in labels and use labels to supervise the learning of hash codes. Flexible Discrete Multi-view Hashing (FDMH)Liu et al. (2020) proposes a collaborative learning strategy that encodes visual and semantic embeddings into a consistent Hamming space. With the development of deep learning, some works have applied deep learning methods to supervised MMH tasks. Deep Collaborative Multi-View Hashing (DCMVH)Zhu et al. (2020) is the first to propose a deep MMH model. It designs a fusion network that deeply integrates multi-modal features and learns representations that include low-level multi-view feature distributions and high-level semantics. Inspired by the success of graph convolutional networks (GCNs)Kipf & Welling (2016), Flexible Graph Convolutional Multi-modal Hashing (FGCMH)Lu et al. (2021a) constructs a graph structure for each modality. Through intra-modal GCNs, it extracts intra-modal structural information. Additionally, hash GCNs and semantic GCNs process the fused graph after modality fusion.

## 2.2 PARTIAL MULTI-MODAL HASHING

To address the issue of incomplete modality data in practice, several attempts have been made. For cross-modal scenarios, Partial Multi-Modal Hashing (PM$^2$H) Wang et al. (2015) was the first to attempt partial cross-modal hashing. It utilizes complete modalities to explore cross-modal consistency and enhances the representation of hash codes through orthogonal rotation. Semi-Paired Discrete Hashing (SPDH) Shen et al. (2017) uses fully paired anchors to leverage partial samples, constructing a common subspace for both complete and incomplete samples. The Collective Affinity Learning Method (CALM) Guo & Zhu (2020) is an unsupervised method that uses collective affinity reconstruction to learn anchor graphs for each modality. It also proposes adaptive affine fusion to adaptively fuse adjacency information from different modalities. Incomplete Cross-Modal Retrieval with Deep Correlation Transfer (ICMR-DCT) Shi et al. (2024) leverages available modalities and neighboring relationships in partial multi-modal samples. It uses a graph attention network (GAT)Veličković et al. (2018)-based encoder to generate missing modalities and compress multimodal features into a subspace.

These cross-modal retrieval methods for handling missing modalities are inspiring, but they cannot be directly applied to partial MMH tasks. FOMH Lu et al. (2019) and FGCMH Lu et al. (2021b) propose flexible MMH models that can handle missing modalities during the query phase of multimodal retrieval but cannot utilize partial multi-modal samples during the training phase. SAPMH Zheng et al. (2021), GCIMH Shen et al. (2023), and NCH Tan et al. (2023) can handle partial multi-modal samples during both training and query phases. SAPMH proposes a shallow hashing method to construct latent representations for each available modality; however, shallow methods lack semantic representation capabilities compared to deep learning frameworks. GCIMH designs a teacher-student structure with three modules. It first uses mean imputation to complete missing modalities, then inputs the completed data into two teacher networks to extract information. However, simply using fixed values for imputation may introduce erroneous information, misleading subsequent processing. NCH adopts a Transformer Encoder Vaswani et al. (2017) structure to generate missing modality data through neighboring samples with the same label, ignoring the discriminative information between samples of different tcategories. Additionally, NCH can only use fully paired anchors to generate missing modality data. Considering these issues, we propose a method named JCPMH to effectively complete missing modalities and generate robust hash codes.

## 3 THE PROPOSED METHOD

In this section, we will introduce the proposed method.

### 3.1 PROBLEM DEFINITION

The goal of the proposed JCPMH is to complete the partial multi-modal samples and then learn a collection of compact $P$-bit hash code $B \in \{-1, 1\}^{N \times P}$ for these completed samples and fully-paired samples. To enhance the performance of the downstream tasks of multi-modal retrieval, we must ensure that the learned hash codes can compress sufficient multi-modal information from the original data.

In order to compare with other related works on some datasets, our work mainly focuses on two modalities, typically image and text modality. Given the incomplete training set $\mathbb{O} = \{\mathbb{I}, \tilde{\mathbb{I}}_1, \tilde{\mathbb{I}}_2\}$ with N samples, where $\mathbb{I} = \{X_n^c, Y_n^c, L_n^c\}_{n=1}^{N_1}$ means fully-paired samples. $\tilde{\mathbb{I}}_1 = \{X_n^i, \circ, L_n^i\}_{n=1}^{N_2}$ and $\tilde{\mathbb{I}}_2 = \{\circ, Y_n^t, L_n^t\}_{n=1}^{N_3}$ are partial multi-modal samples with only image or text modality respectively. In addition, $N = N_1 + N_2 + N_3$. Suppose that we have $K$ classes and label vector $L_n^* \in \{0, 1\}^{1 \times K}$. $L_{ni}^* = 1$ indicates the $n$-th sample can be divided into $i$-th category, otherwise $L_{ni}^* = 0$.

As show in Figure.1, Our method mainly consists of three basic parts: a classification module, an autoencoder and a multi-modal hashing network. The autoencoder and classification module jointguide the generator to complete the missing modality in partial multi-modal samples. This data is then fed into the hashing network to generate a compact representation.

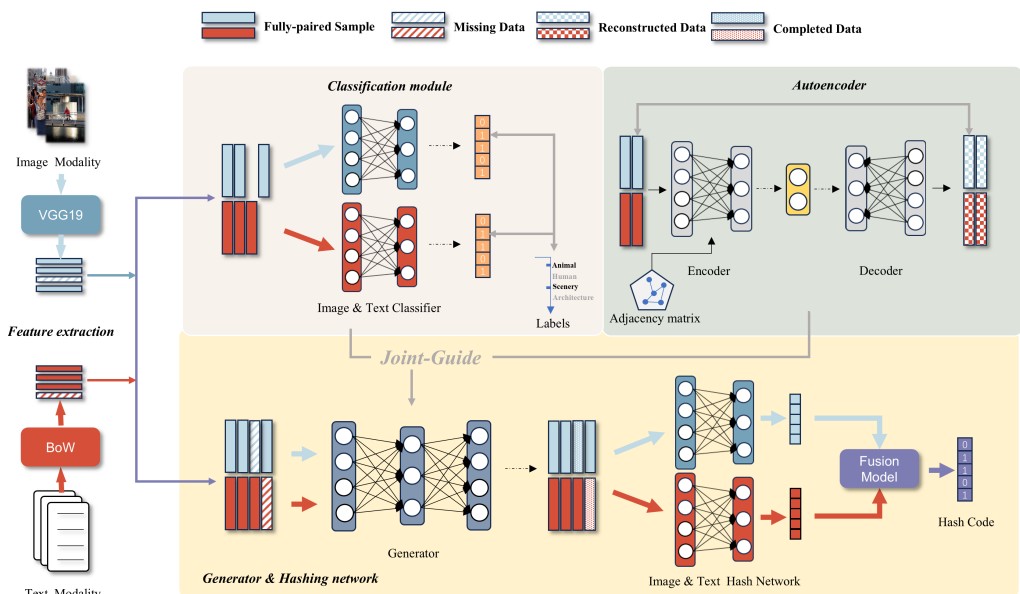

Figure 1: The framework of our proposed JCPMH. The features extracted from the original image and text modalities serve as the input to the network. Network consists of three modules: Classification Module, Autoencoder and the Hashing Network. Classification Module and Autoencoder extract information from the available samples and joint-guided generator to generate missing modality data.

## 3.2 CLASSIFICATION MODULE

In order to learn the discriminative information between samples of different labels, we designed the classification module.

Specifically, we trained a classifier for each modality. As show in Figure.1, we use the available samples of each modality as training samples, which represents as $[X^c, X^i]$ and $[Y^c, Y^t]$.Take the classifier of image modality as an example. The outputs of the classifier can be represents as $\hat{L} = f_{Ci}([X^c, X^i]; \Theta_{Ci})$, where $\Theta_{Ci}$ denotes the parameter of classifier. And the ground truth label $L = [L_n^c, L_n^i]$, then we can train the classifier by optimizing the loss $\mathcal{L}_c$:

$$\mathcal{L}_c = -\frac{1}{N_1 + N_2} \sum_{n=1}^{N_1+N_2} L \log \hat{L} + (1 - L) \log (1 - \hat{L}) \tag{1}$$

Note that, a sample may belong to multiple categories, so this is a multi-label classification task. The process of training the classifier for text modality $f_{Ct}([Y^c, Y^i]; \Theta_{Ct})$ is similar.

Assuming that our total sample has a PDR of 70%, which means there are only 30% of the samples as fully-paired samples, and the other 70% of samples are partial multi-modal samples with only image or text modality respectively. If we only use the structure of the autoencoder, we can only utilize 30% of the samples. However, by using the classification module, we can utilize the remaining partial multi-modal samples. So in another way, this module has improved the utilization of samples.

## 3.3 AUTOENCODER

We believe that there is rich structural information in multi-modal samples, not only in a single modality, but also implicit structural information above paired modalities. To extract this overall structural information, we design an autoencoder, concatenate fully-paired samples which combine the original image and text features $X^c$ and $Y^c$. And then fed them into the autoencoder:

$$[\hat{X}^c, \hat{Y}^c] = f_{Ai}([X^c, Y^c], A^c; \Theta_A), \tag{2}$$

where $\hat{X}_n^c$ and $\hat{Y}_n^c$ denote the image and text reconstruction vector output by the autoencoder, and $\Theta_A$ denotes the parameter of autoencoder. In addition to this, to extract information from the labels,

we also introduced the structure of the adjacency matrix $A^c \in \mathbb{R}^{N_1 \times N_1}$ as $A^c = L^c(L^c)^T$ into the forward propagation process of the autoencoder:

$$H^{(l+1)} = ReLu(\tilde{A}^c H^l W^l) \tag{3}$$

$H^l$ indicates the output of the $l$-th layer, $W^l$ is parameter of the $l$-th layer in autoencoder, $\tilde{A}^c$ comes from $A^c$:

$$\tilde{A}^c_{ij} = \frac{1 - exp(-A^c_{ij})}{1 + exp(-A^c_{ij})} \tag{4}$$

the value of $A^c_{ij}$ reflects the correlation between $i$-th sample and $j$-th sample to a certain extent. In order to map the value of $A^c_{ij}$ to between 0 and 1, while keeping the 0 unchanged, we designed the above process.

In summary, we propose the following loss function $\mathcal{L}_r$ to train the autoencoder:

$$\mathcal{L}_r^i = \frac{1}{N_1} \sum_{n=1}^{N_1} ||\hat{X}_n^c - X_n^c||^2, \tag{5}$$

$$\mathcal{L}_r^t = -\frac{1}{N_1} \sum_{n=1}^{N_1} Y_n^c \log \hat{Y}_n^c + (1 - Y_n^c) \log (1 - \hat{Y}_n^c), \tag{6}$$

$$\mathcal{L}_r = \mathcal{L}_r^i + \mathcal{L}_r^t \tag{7}$$

where $\hat{X}_n^c$ and $\hat{Y}_n^c$ are $n$-th row of $\hat{X}^c \in \mathbb{R}^{N_1 \times d_i}$ and $\hat{Y}^c \in \mathbb{R}^{N_1 \times d_t}$. Considering the structural differences between image and text modalities, we use Mean Squared Error (MSE) and Binary Cross Entropy (BCE) as loss functions respectively.

### 3.4 Multi-Modal Hashing Learning

In order to convert fully-paired samples or samples completed by the generator into compact binary codes, we designed a hashing network as shown in Figure.1. First, we assign a sub-network to the image and text modality respectively to compress them into the same length, and then combine the two through the fusion network to transfer into a representation of a specific length:

$$h = f_u(h_x + h_y; \Theta_u) \tag{8}$$

where $h_x$ and $h_y$ are the text and image modalities that have been compressed, $\Theta_u$ is the parameter of fusion network, $h \in \mathbb{R}^{N \times P}$ is the intermediate representation of the final hash codes $B$:

$$B = sign(h) \tag{9}$$

Similar to the autoencoder mentioned above, we introduce the structure of the adjacency matrix to train the hashing network, in order to extract rich features of different modalities and remain discriminative information:

$$\mathcal{L}_s = \frac{1}{N} ||h^T h - A||^2 \tag{10}$$

$$\mathcal{L}_b = \frac{1}{N} ||h - B||^2 \tag{11}$$

where $A \in \mathbb{R}^{N \times N}$ is a similarity matrix similar to $A_c$ mentioned above, which represent the label similarity between samples. Moreover, in order to control the quantization error caused by the sign function in Eq.9, we adopted the quantization loss $\mathcal{L}_b$.

### 3.5 Objective Function

After training the two modules mentioned above, we need to transfer the information from them to the generator. On one hand, to ensure that the completion information output by the generator contains sufficient multi-modal global structural information from the autoencoder, we designed the

---

**Algorithm 1** The learning algorithms for JCPMH.

---

**Require:**
  $\mathbb{O} = \{\mathbb{I}, \tilde{\mathbb{I}}_1, \tilde{\mathbb{I}}_2\}$: Incomplete training set;
     $P$: Hash code length;
     $\alpha, \lambda_1, \lambda_2$: Hyper-parameters;
**Ensure:**
  $B$: Multi-modal hash codes
  **Initialization:** Epoch for training Classification module, Autoencoder and Hashing network $T_1$,
  $T_2, T_3$;
1: **for** $i = 1 : T_1$ **do**
2:     Update $\Theta_{Ai}$ and $\Theta_{At}$ via Eq.7
3: **end for**
4: **for** $i = 1 : T_2$ **do**
5:     Update $\Theta_{Ci}$ and $\Theta_{Ct}$ via Eq.1
6: **end for**
7: **for** $i = 1 : T_1$ **do**
8:     Fixed the parameters of teacher networks above and jointly update $\Theta_g$ and $\Theta_u$ via Eq.14
9: **end for**

---

Eq.12. On the other hand, as depicted in Eq.13, we use a cross-entropy loss to get discriminative information extracted by the classification module:

$$\mathcal{L}_2 = \frac{1}{N}||[\hat{X}^*, \hat{Y}^*] - [X^*, Y^*]||^2 \tag{12}$$

$$\mathcal{L}_3 = \frac{1}{N}(L^* log \hat{L}^* + (1 - L^*) log(1 - \hat{L}^*)) \tag{13}$$

where $X^*$ and $Y^*$ are fully-paired samples or samples completed by the cross-modal generator $f_g(I_p; \Theta_g)$, which taking partial modality $I_p$ as input, then generate the corresponding missing modality data. $L^*$ are their labels. Correspondingly, $\hat{X}^*$, $\hat{Y}^*$, and $\hat{L}^*$ represent their respective outputs after passing through the autoencoder or classification module.

Finally, we present the following total objective function:

$$\mathcal{L} = \mathcal{L}_1 + \lambda_1\mathcal{L}_2 + \lambda_2\mathcal{L}_3 \tag{14}$$

where $\mathcal{L}_1 = \mathcal{L}_s + \alpha\mathcal{L}_b$, $\alpha, \lambda_1, \lambda_2$, are hyper-parameters to balance each term. The overall training process is provided in Alg.1.

## 4 EXPERIMENTS

### 4.1 DATASETS

In order to compare with models performing the same tasks, we have chosen the MIRFLICKR-25K and NUS-WIDE public datasets. These two large-scale multi-modal (visual and language) datasets are commonly used as benchmark datasets in multi-modal learning and multi-modal retrieval research. Visual features are extracted using VGGNet, while textual features are represented by Bag-of-Words (BoW) vectors. The details of these two datasets are as follows:

**MIR Flickr** consists of 25,000 images collected from the social photography website Flickr, each accompanied by 24 annotations. We extracted 20,015 samples from this dataset. Of these, 2,243 pairs of samples are used as the query set, while the remaining 17,772 pairs form the database. Additionally, 5,000 pairs of samples are randomly selected for training. As previously mentioned, we use the extracted features, where each pair of samples includes a 4096-dimensional visual feature and a 1386-dimensional text vector.

**NUS-WIDE** dataset contains 269,648 images selected from the web along with corresponding tags. From this, we selected 195,834 pairs of image-text samples. These samples belong to the 21 most frequent categories. We selected 2,085 samples as the query set, and the remaining 193,749 samples

serve as the database, a random subset of 21,000 samples from the database is used for training. Similarly, the image features and text features we use are vectors of 4096 dimensions and 1000 dimensions respectively.

## 4.2 Experimental Setting and Evaluation Metric

Our experiment consists of five parts. The first part is conducted on fully-paired samples, which can be considered a traditional multimodal hashing task. We will compare our method with the following multimodal hashing models: DMVH, SDMH, FOMH, FDMH, DCMVH, SAPMH, FGCMH, NCH, GCIMH. The second part, which is also crucial, involves conducting experiments in a scenario with missing modalities. In this context, only the SAPMH, NCH, and GCIMH models mentioned above are capable of handling missing modalities during both the training and query stages.

Additionally, we conducted a series of experiments to evaluate the effectiveness of missing data completion in subsection 4.5. In subsection 4.6 and subsection 4.7, we conducted a detailed analysis of the results from ablation studies and parameter sensitivity analysis.

For evaluation metrics, mAP (mean Average Precision) serves as our main metric. It is widely used in image retrieval and multi-modal retrieval tasks. mAP requires the participation of all samples from the database in its calculation, making it an effective reflection of the retrieval capability of hash codes.

We run the model on a 16GB VRAM V100 GPU. For the classification module, both the image and text classifiers are MLPs. The hidden layers have dimensions of 2048 and 512, respectively. The output layer length corresponds to the number of categories: 24 for the Flickr dataset and 21 for the NUS-WIDE dataset. For the autoencoder, its encoder consists of two layers of Graph Convolutional Networks (GCN) with an intermediate hidden layer of 4096 dimensions. The decoder comprises two linear layers that separately output reconstructed image and text data, using ReLU as the activation function. For the hashing network, we designed two MLPs as generators. Each takes one modality (either image or text) as input and generates the other modality. The hidden layer dimensions are 2048. Similarly, the hashing networks for both image and text are also MLPs, transforming the incomplete text and image features into a unified 1024-dimensional representation. Finally, the fusion model is a linear layer, and its output dimension corresponds to the final hash code length. Besides, we set the hyper-parameters $\{\alpha = 0.1, \lambda_1 = 0.1, \lambda_2 = 0.25\}$, $\{\alpha = 0.1, \lambda_1 = 0.1, \lambda_2 = 0.05\}$ for MIR Flickr and NUS-WIDE, respectively.

## 4.3 Complete Multi-modal Retrieval

We compare JCPMH with other methods on fully-paired samples. However, considering that most modules in our model will not function when dealing with fully-paired samples, it degenerates into a vanilla hashing network. In order to fully activate the model, we set a small PDR for JCPMH, which is 10% on both the training set and the query set. For fair comparison, we have applied the same treatment to other models that can handle partial MMH tasks. The experimental results are shown in the Table.1. Under different experimental settings, our model surpasses the best-performing traditional MMH model FGCMH by an average of 4.1%, and surpasses the second-best partial MMH method NCH by an average of 0.7%.

## 4.4 Partial Multi-modal Retrieval

In this section, we analyze the most critical experiments. As shwon in Table.2, we conducted experiments under three different scenarios: when the training set is incomplete, when the query set is incomplete, and when both are incomplete. The PDR is uniformly set to 70%. From the mAP results, it can be seen that our method can handle different situations well and shows improvement over other models under various experimental settings. For example, when the PDR of both the training set and the query set is set to 70%, considering different hash code lengths, our model shows an average improvement of 1.37% compared to the second-best method NCH on the NUS-WIDE dataset. In addition, we analyze the results of JCPMH with the variations in hash code length and PDR. As shown in Figure.2, with the hash code length increases, our mAP also continues to improve. It can be seen that our method outperforms both GCIMH and SAPMH. When the hash code length is relatively small, the improvement of our model compared to NCH is more significant,

Table 1: mAPs of different models training and testing on fully-paired samples

| Methods | MIR Flickr | | | | NUS-WIDE | | | |
|---|---|---|---|---|---|---|---|---|
| | 16bits | 32bits | 64bits | 128bits | 16bits | 32bits | 64bits | 128bits |
| DMVH | 0.7231 | 0.7326 | 0.7495 | 0.7641 | 0.5676 | 0.5883 | 0.6092 | 0.6279 |
| SDMH | 0.7316 | 0.7400 | 0.7568 | 0.7723 | 0.6321 | 0.6346 | 0.6626 | 0.6648 |
| FOMH | 0.7557 | 0.7632 | 0.7654 | 0.7705 | 0.6329 | 0.6456 | 0.6678 | 0.6791 |
| FDMH | 0.7802 | 0.7963 | 0.8094 | 0.8181 | 0.6575 | 0.6665 | 0.6712 | 0.6823 |
| DCMVH | 0.8097 | 0.8279 | 0.8354 | 0.8467 | 0.6509 | 0.6625 | 0.6905 | 0.7023 |
| SAPMH* | 0.7676 | 0.7939 | 0.8022 | 0.8101 | 0.6272 | 0.6644 | 0.6733 | 0.6852 |
| FGCMH | 0.8173 | 0.8358 | 0.8377 | 0.8406 | 0.6677 | 0.6874 | 0.6936 | 0.7011 |
| GCIMH* | 0.8169 | 0.8318 | 0.8355 | 0.8430 | 0.6416 | 0.6621 | 0.6894 | 0.7072 |
| NCH* | 0.8211 | 0.8409 | 0.8527 | 0.8570 | 0.7126 | 0.7360 | 0.7578 | 0.7710 |
| JCPMH*(ours) | **0.8278** | **0.8483** | **0.8587** | **0.8606** | **0.7224** | **0.7488** | **0.7617** | **0.7772** |

[1] The bold data represents the best results, while the underlined data corresponds to the second-best results.

[2] The models marked with (*) obtained results under the condition that the training set and query set have a PDR of 10%.

Table 2: mAPs of different models training and testing on partial multi-modal samples

| Methods | MIR Flickr | | | | NUS-WIDE | | | |
|---|---|---|---|---|---|---|---|---|
| | 16bits | 32bits | 64bits | 128bits | 16bits | 32bits | 64bits | 128bits |
| SAPMH | 0.7305 | 0.7586 | 0.7718 | 0.7800 | 0.6001 | 0.6209 | 0.6305 | 0.6446 |
| GCIMH | 0.8114 | 0.8274 | 0.8311 | 0.8233 | 0.6028 | 0.6236 | 0.6491 | 0.6502 |
| NCH | 0.8119 | 0.8300 | 0.8458 | 0.8513 | 0.7060 | 0.7315 | 0.7520 | 0.7669 |
| JCPMH(ours) | **0.8237** | **0.8403** | **0.8535** | **0.8552** | **0.7158** | **0.7401** | **0.7539** | **0.7687** |
| SAPMH | 0.7359 | 0.7741 | 0.7831 | 0.7909 | 0.5382 | 0.5490 | 0.5696 | 0.5648 |
| GCIMH | 0.7620 | 0.7899 | 0.8127 | 0.7949 | 0.5586 | 0.5789 | 0.5996 | 0.6024 |
| NCH | 0.7920 | 0.8109 | 0.8170 | 0.8222 | 0.6970 | 0.7100 | 0.7252 | 0.7408 |
| JCPMH(ours) | **0.8026** | **0.8184** | **0.8299** | **0.8305** | **0.7020** | **0.7206** | **0.7337** | **0.7433** |
| SAPMH | 0.7222 | 0.7462 | 0.7533 | 0.7570 | 0.5327 | 0.5471 | 0.5708 | 0.5735 |
| GCIMH | 0.7892 | 0.7997 | 0.7958 | 0.8101 | 0.5648 | 0.5721 | 0.5931 | 0.6146 |
| NCH | 0.7915 | 0.8050 | 0.8207 | 0.8235 | 0.6820 | 0.7030 | 0.7224 | 0.7358 |
| JCPMH(ours) | **0.7993** | **0.8134** | **0.8260** | **0.8281** | **0.6980** | **0.7268** | **0.7331** | **0.7402** |

[1] The three sections in the table, from top to bottom, represent three different experimental settings: PDR of 70% for only training set; PDR of 70% for only query set; both the training set and query set share a PDR of 70%.

which reflects the ability of JCPMH to extract and compress modality information. As shown in Figure 3, our results show a slow decline as the PDR increases. Even at a PDR of 90%, we can still obtain relatively good results, while the results of GCIMH and SAPMH show some fluctuations, demonstrating the stability of our method..

## 4.5 EFFECTIVENESS EVALUATE

Our method focuses on the completion of missing modalities. To demonstrate the effectiveness of the completion process, we conducted a visualization experiment on the NUS-WIDE dataset. As shown in Figure 4, we have selected several other completion strategies for comparison.

For the strategy of imputing with zeros, a considerable amount of modal information is lost, and inaccurately completed samples may mislead the generation of hash codes. As seen in Figure 4.a, this strategy lacks effectiveness.

For the strategy of imputing with the mean value (generated by neighboring samples with the same label), which is adopted by GCIMH, the completed samples tend to be concentrated, as reflected in Figure 4.b. The uniform completion content increases the homogeneity between samples, ignoring

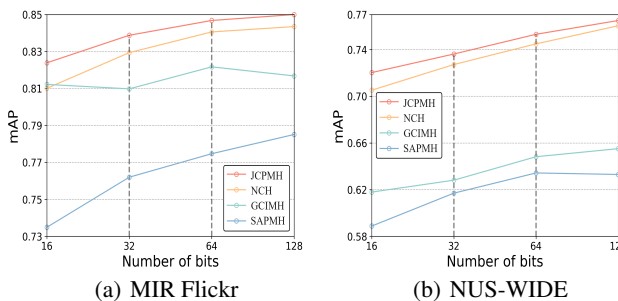

Figure 2: mAPs with respect to the number of bits on MIR Flickr and NUS-WIDE when PDR of training set and query set is fixed at 30%

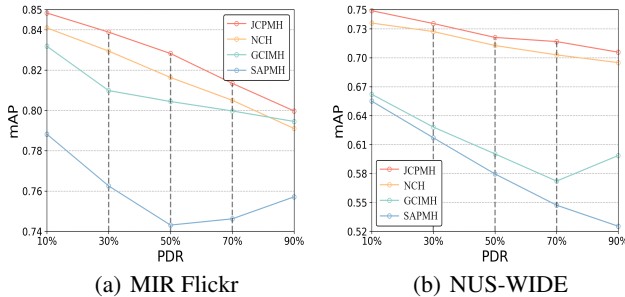

Figure 3: Variations of PDR with the change of PDR, the hash code length is fixed at 32 bits

the differences between them.

For NCH, it introduces the information of neighbors for completion. However, the distribution of the completed data does not restore the original data well, lacking the discriminative information of data belonging to the same category.

In contrast, JCPMH, as shown in Figure 4.d, simulates the distribution of fully-paired samples well, preserving the rich structural and discriminative information of modalities. This demonstrates the effectiveness of JCPMH in completing missing modalities.

### 4.6 ABLATION STUDY

JCPMH can function with either the autoencoder or the classification module as the guide module. Therefore, we set up two variants: JCPMH-A, with the autoencoder removed, and JCPMH-B, with the classification module removed. The results of the ablation experiment can be seen in Table3. It can be observed that the joint operation of both modules effectively enhances performance.

Table 3: Results of ablation study

| Methods | MIR Flickr | | | | NUS-WIDE | | | |
|---------|------------|--------|--------|---------|----------|--------|--------|---------|
| | 16bits | 32bits | 64bits | 128bits | 16bits | 32bits | 64bits | 128bits |
| JCPMH-A | 0.8039 | 0.8119 | 0.8237 | 0.8357 | 0.6732 | 0.7049 | 0.7210 | 0.7321 |
| JCPMH-B | 0.8048 | 0.8182 | 0.8298 | 0.8360 | 0.6866 | 0.7054 | 0.7308 | 0.7354 |
| JCPMH | 0.8116 | 0.8282 | 0.8374 | 0.8414 | 0.7012 | 0.7211 | 0.7386 | 0.7507 |

[1] JCPMH-A and JCPMH-B are variations of JCPMH with Grah Auto-encoder and Classification module removed, respectively.

[2] We set PDR of 50% for both training set and query set.

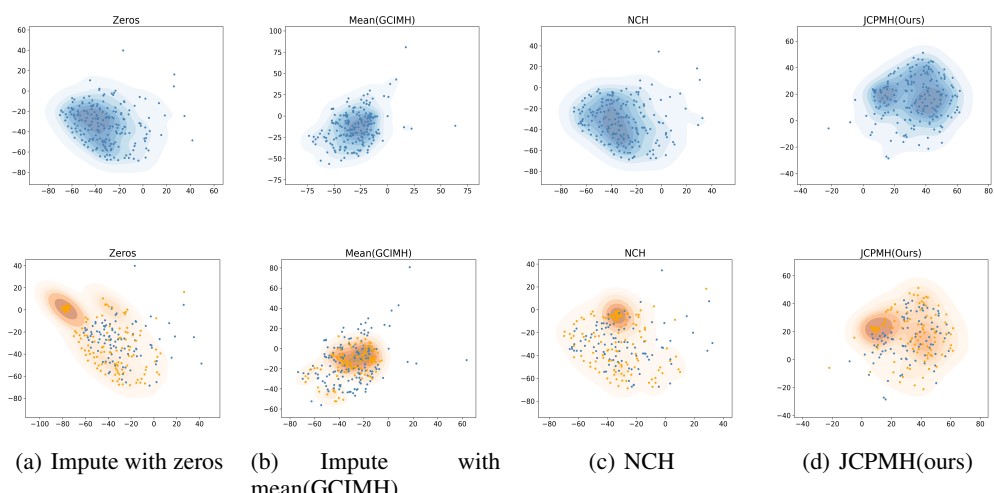

(a) Impute with zeros    (b)    Impute    with    (c) NCH    (d) JCPMH(ours)
                             mean(GCIMH)

Figure 4: Visualization of imputed data on NUS-WIDE datasets. We randomly selected samples belonging to the same label and set half of them missing visual modality while the other half to be missing text modality. The figure in blue shows the distribution of original fully-paired data and orange data points below are samples completed by different methods.

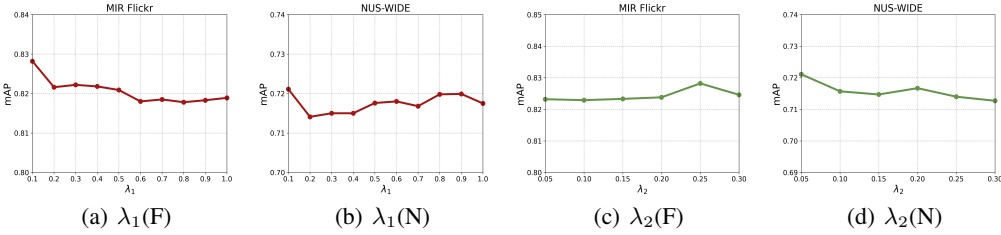

(a) $\lambda_1$(F)    (b) $\lambda_1$(N)    (c) $\lambda_2$(F)    (d) $\lambda_2$(N)

Figure 5: Parameter sensitivity curves of JCPMH on MIR Flickr and NUS-WIDE with fixed hash code length and PDR.

### 4.7 PARAMETER SENSITIVITY ANALYSIS

We selected two important hyper-parameters, $\lambda_1$ and $\lambda_2$, for Parameter Sensitivity Analysis. These two hyper-parameters represent the influence of the two guide modules, the Autoencoder and the Classification Module, on the generator. We analyze each parameter while keeping the other one fixed. The variation curve is shown in Figure 5. It can be seen that as the hyper-parameters change, the results maintain a small range of variation, reflecting the stability and robustness of JCPMH.

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
