# OpenReview forum: "Autoencoder and Classifier based Joint-Guided Completion for Partial Multi-Modal Hashing"
_ICLR.cc/2025/Conference — ICLR 2025 Conference Withdrawn Submission_

### Official Review · Reviewer_YQX2 · 2024-10-15

**Soundness:** 2
**Presentation:** 2
**Contribution:** 2
**Rating:** 3
**Confidence:** 4

**Summary:**

The paper titled "Autoencoder and Classifier Based Joint-Guided Completion for Partial Multi-Modal Hashing" proposes a novel method called JCPMH (Joint-Guided Completion for Partial Multi-Modal Hashing). The primary goal of JCPMH is to address the challenge of incomplete multi-modal samples in multi-modal hashing (MMH) tasks. Traditional MMH methods struggle with partial multi-modal data, often relying on fully-paired samples. JCPMH integrates autoencoders and classifiers to effectively handle incomplete multi-modal samples by generating accurate completion information.

**Strengths:**

*Effective Handling of Incomplete Data*: JCPMH can effectively handle partial multi-modal samples, which is a common issue in real-world datasets.

*Comprehensive Experiments*: The paper includes extensive experiments on multiple datasets, and provides detailed analyses of the results.

*Presentation*: Good English writing, easy to follow.

**Weaknesses:**

*Hyperparameter Sensitivity*: The performance of JCPMH might be sensitive to the choice of hyperparameters, which could require careful tuning.

*Incomplete paper*: Lack the section of conclusion.

*Weak motivation*: The assumption is weak. In reality, when an incomplete problem occured, it tends to be incomplete in both train and test datasets, thus global fully-paired samples will not exist.

*Weak baseline*: Baselines are weak and outdated. More recent work should be cited.

**Questions:**

Overall, the paper is not ready to submit as a ICLR paper, and experiment is not convincing, motivation is not solid. The conclusion section is important for any paper.

Reviewer's recommandation is reject.

---

### Official Review · Reviewer_YDav · 2024-10-27

**Soundness:** 2
**Presentation:** 2
**Contribution:** 2
**Rating:** 3
**Confidence:** 5

**Summary:**

This paper proposes an autoencoder and classifier based joint-guided completion method for partial multi-modal hashing, called JCPMH. It completes the missing modalities during the training and test stages. It first trains the multi-modal classifiers and a global autoencoder as teachers, and then use them to guide the missing features generation. The completed multi-modal features are fused to produce the hash codes.  The authors conduct experiments to two benckmarks to evaluate the method.

**Strengths:**

1.	The motivations are clear, and the paper is easy to read.
2.	The experiments on Flickr and NUSWIDE datasets show the effectiveness of the method under various partial data rates.

**Weaknesses:**

1.	The main idea is to pre-train a classifier and autoencoder to guide the missing feature generation. While it is a practical and likely effective solution, it is straightforward and fails to provide a fresh perspective. Each module is very common, and the pipeline of hash codes learning is also widely-used. The overall novelty of this paper does not meet the standard required for ICLR.
2.	The proposed method cannot handle the case when there is no paired data (PDR = 1), because the pre-trained autoencoder requires complete multi-modal data.
3.	The paper is incomplete without Conclusion.
4.	Some minor issues, for example, at line 42, “samplesa”; Captions of figure 3, “Variations of PDR”.

**Questions:**

See weaknesses.

---

### Official Review · Reviewer_Joza · 2024-10-27

**Soundness:** 2
**Presentation:** 1
**Contribution:** 1
**Rating:** 3
**Confidence:** 5

**Summary:**

This manuscript introduces a partial multi-modal hashing method named JCPMH. The method focuses on recover partial multi-modal instances through a missing feature generator, which is guided by a discriminative single-modal classifier and a multi-modal autoencoder learned from present samples on a prior stage. Experiments validate the effectiveness of the method.

**Strengths:**

The proposed method performs knowledge distillation from characteristics of seen samples to the missing feature generator, introducing an interesting learning scheme for missing modality recovery.

**Weaknesses:**

1. Unclear general motivation. More explanation of the joint-guide learning mode is required.
2. Careless list of related works. In fact, the work [1] around Line 112 solves multi-modal hashing instead of only cross-modal hashing.
3. Weak method novelty. The learning scheme of guided generation of partial data is also seen in the compared method NCH and other methods including partial cross-modal hashing methods [2-3]. Considering these works, the manuscript displays very limited novelty.
4. Ambiguous notations. Vectors and matrices are not distinguished from scalars.
5. Incremental contribution. Experimental results only show marginal improvements. Meanwhile, the ablation study only compare variants removing the classifier or the autoencoder, not touching the core contribution of the two-step guided recovery mode, i.e., variants without completion or jointly training all components.
6. Unclear figures. Figure 4 has no explanation of its components, making the comparison not understandable. Meanwhile, using different data distribution across methods also induces ambiguity.
7. Low readability. Many language mistakes and typos are present in the manuscript. Meanwhile, there is no conclusion and limitation sections.

[1] Wang, Q., Si, L., & Shen, B. (2015, June). Learning to hash on partial multi-modal data. In Twenty-Fourth International Joint Conference on Artificial Intelligence.

[2] Jing, M., Li, J., Zhu, L., Lu, K., Yang, Y., & Huang, Z. (2020, October). Incomplete cross-modal retrieval with dual-aligned variational autoencoders. In Proceedings of the 28th ACM International Conference on Multimedia (pp. 3283-3291).

[3] Zeng, Z., Wang, S., Xu, N., & Mao, W. (2021, July). Pan: Prototype-based adaptive network for robust cross-modal retrieval. In Proceedings of the 44th international ACM SIGIR conference on research and development in information retrieval (pp. 1125-1134).

**Questions:**

1. The authors claim that directly using partial data often leads to loss of multi-modal semantic information. However, the completed data are also generated from the seen dataset. Why can this simple recovery solve semantic loss?
2. Why not include more effective ablations for the core contribution? The current results are not convincing.

---

### Official Review · Reviewer_rmi9 · 2024-10-31

**Soundness:** 3
**Presentation:** 3
**Contribution:** 2
**Rating:** 5
**Confidence:** 5

**Summary:**

In this paper, the authors present a method for partial multi-modal hashing task. Specifically, the authors design a multi-modal classification module to learn different information. In addition, multi-modal data are concatenated into a whole through an autoencoder. Based on the above modules, a generator is learned to generate completion information. The results demonstrate the effectiveness of the proposed method.

**Strengths:**

1. The idea of using a classification module and an autoencoder to guide a generator is technically sound.
2. The paper is clearly written. It is easy to follow the idea of this paper.
3. As shown in the experiments, the proposed method is able to achieve better results than SOTAs.

**Weaknesses:**

1. More details about the algorithm or the experiments should be given.
2. Some further analysis on the performance of the proposed model should be conducted.
3. The authors just concatenate different modalities as the input of the autoencoder. It may be not a most effective scheme.
3. There are some English problems of the writing.

**Questions:**

1. The authors update both the classification module and the hashing network with T1 iterations. Why? In addition, how to determine the value the T1, T1, etc.
2. In the proposed method, the generator is guided by the classification module and the autoencoder. It means that the performance of the proposed model highly depends on the classification module and the autoencoder. There is no further analysis on this.
3. What data are used to train the classification module and the autoencoder?

---

### Note · Authors · 2024-11-18

I have read and agree with the venue's withdrawal policy on behalf of myself and my co-authors.